# Does Colchicine Substitute Corticosteroids in Treatment of Idiopathic and Viral Pediatric Pericarditis?

**DOI:** 10.3390/medicina55100609

**Published:** 2019-09-20

**Authors:** Vladislav Vukomanovic, Sergej Prijic, Stasa Krasic, Ruzica Borovic, Sanja Ninic, Dejan Nesic, Bojko Bjelakovic, Sasa Popovic, Mila Stajević, Gordana Petrović

**Affiliations:** 1Cardiology Department, Mother and Child Health Care Institute of Serbia “Dr.Vukan Cupic”, 11070 Belgrade, Serbia; sergej.prijic@med.bg.ac.rs (S.P.); ds185097@sudent.mfub.bg.ac.rs (S.K.); Sanja.ninic@med.bg.ac.rs (S.N.); ds185133@sudent.mfub.bg.ac.rs (S.P.); 2School of Medicine, University of Belgrade, 11000 Belgrade, Serbia; dejan.nesic@med.bg.ac.rs (D.N.); mila.stajevic@med.bg.ac.rs (M.S.); 3Pediatrics Department, Hospital “Sveti Vracevi”, 76300 Bijeljina, Bosnia and Herzegovina; borovicruzica@gmail.com; 4Institute of Medical Physiology “Rihard Burian”, 11000 Belgrade, Serbia; 5Clinic of Pediatrics, Clinical Center Nis, School of Medicine, University of Nis, 18000 Nis, Serbia; bojko967@gmail.com; 6Cardiac Surgery Department, Mother and Child Health Care Institute of Serbia “Dr.Vukan Cupic”, 11070 Belgrade, Serbia; 7Immunology Department, Mother and Child Health Care Institute of Serbia “Dr.Vukan Cupic”, 11070 Belgrade, Serbia; gordana.petrovic.im@gmail.com

**Keywords:** treatment, pericarditis, colchicine, corticosteroid, childhood

## Abstract

*Background and Objectives*: Recurrence of pericarditis (ROP) is an important complication of the acute pericarditis. The aim of this study was to analyse the influence of aetiology, clinical findings and treatment on the outcome of acute pericarditis. *Methods*: Data were retrospectively collected from medical records of patients treated from 2011 to 2019 at a tertiary referent heart paediatric center. *Results*: Our investigation included 56 children with idiopathic and viral pericarditis. Relapse was registered in 8/56 patients, 2/29 (7.41%) treated with nonsteroidal anti-inflammatory drugs (NSAID) and 6/27 (28.57%) treated with corticosteroids (CS) and NSAID. Independent risk factors for ROP were viral pericarditis (*p* = 0.01, OR 31.46), lack of myocardial affection (*p* = 0.03, OR 29.15), CS use (*p* = 0.02, OR 29.02) and ESR ≥ 50 mm/h (*p* = 0.03, OR 25.23). In 4/8 patients the first recurrence was treated with NSAID and colchicine, while treatment of 4/8 patients included CS. Children with ROP treated with CS had higher median number of recurrence (5, IQR: 2–15) than those treated with colchicine (0, IQR: 0–0.75). *Conclusions*: Independent risk factors for recurrence are CS treatment, viral aetiology, pericarditis only and ESR ≥ 50 mm/h. Acute pericarditis should be treated with NSAID. Colchicine and NSAID might be recommended in children with the first ROP.

## 1. Background

Pericarditis is a common disease in children and the cause of 1–5% of chest pain in childhood [1]. Up to 80–90% of cases of pericarditis are either idiopathic or of viral origin [2]. However, the majority of cases with diagnosed idiopathic acute pericarditis have unrecognized viral or immune aetiology [3,4,5]. Recurrent pericarditis is an important complication of the acute form of the disease. Nevertheless, unlike in adults, there is no consistent data on the incidence and aetiology of both acute and recurrent pericarditis in children. The incidence of recurrence is estimated to be 15–30% in adult patients [6]. The European Society of Cardiology (ESC) has published recommendations and guidelines for pericarditis treatment in adults [7], but standardized therapy protocol for pericarditis in children is yet to be established [8]. There is still a doubt regarding the use of corticosteroids (CS), whereas recently published studies support the positive role of colchicine and anakinra in the treatment of pericarditis [9].

The aim of our study was to analyze the outcome of the idiopathic and viral pediatric pericarditis using different treatment protocols.

## 2. Methods

A retrospective cohort study was carried out at the Mother and Child Health Care Institute of Serbia during the period of January 2011 to March 2019 (Ethical code number 8/34, date of approval 16 May 2019).

The study included all children with idiopathic and viral pericarditis treated in our hospital. Patients were admitted at the institute within a timespan of 24 h after the onset of the disease from regional secondary hospitals and primary healthcare centers. Patients with other aetiologies of the acute pericarditis were excluded from our investigation.

Pericarditis was diagnosed based on the existence of two out of four criteria: (1) chest pain, (2) changes in the electrocardiogram, (3) elevated levels of the acute inflammation parameters (erythrocytes sedimentation rate (ESR), C-reactive protein (CRP), number of white blood cell (WBC) count, and (4) echocardiographic visualization of pericardial effusion [10]. Recurrence of pericarditis (ROP) is diagnosed with a documented first episode of acute pericarditis (index attack), a symptom-free interval of 4–6 weeks or longer and evidence of subsequent recurrence of pericarditis. The following parameters were analyzed: gender, age, acute inflammatory parameters, levels of specific cardiac enzymes and proteins, echocardiographic findings and therapeutic protocol. Bacteriological testing (blood, throat, stool and pericardial fluid culture) was carried out to determine the aetiology of the disease. Viral aetiology was based on the detection of the viral nucleic acids of the most common cardiotropic viruses with the polymerase chain reaction (PCR) technique in the following samples: blood, pericardial fluid, pharyngeal swab and stool. Idiopathic pericarditis was defined as virus negative and without specific origin after a diagnostic workup.

The diagnosis of myocarditis was based on the presence of one or more clinical symptoms (e.g., thoracic pain, dyspnoea, fatigue, palpitations and unexplained cardiogenetic shock and one or more diagnostic criteria (increased level of specific cardiac enzymes and proteins, newly abnormal 12 lead electrocardiogram (ECG), Holter or stress test, decreased left ventricle contractility registered by echocardiography, as well as the finding of oedema in cardiac magnetic resonance) [11,12]. When patients were asymptomatic two or more diagnostic criteria had to exist [12]. A diagnosis of myopericarditis was established in patients with signs and symptoms of acute pericarditis with elevated biomarkers of cardiac injury and with normal function of the left ventricle. However, perimyocarditis was diagnosed in children with acute pericarditis, elevated cardiac biomarkers and segmental or global changes in left ventricle contractility [11].

For the treatment of the acute pericarditis nonsteroidal anti inflammatory drugs (NSAID) and CS were administered, as well as colchicine in patients with ROP. The drugs dosage and tapering were mostly adopted from the ESC guidelines for treatment of pericardial disease [7]. Prior to them, the management of the majority of patients with pericardial disease was the administration of CS in dose of 1 mg/kg. Instead and thereafter we administered CS in a few of patients, and in smaller doses—0.5 mg/kg. Additional, CS was introduced in all patients with affected myocardium, in patients with larger pericardial effusion diameter, after pericardial drainage and in those patients who did not respond adequately with NSAID. 

Corticosteroids were discontinued first and the dose was decreased after the normalization of clinical and laboratory parameters. If the dose had been >50 mg/day, it was decreased to 10 mg/day every 1–2 weeks; if the dose was 25 and 50 mg/day, it was decreased to 5–10 mg/day every 1–2 weeks; when the administered dose was between 15 and 25 mg/day, the decrease was by 2.5 mg/day every 2–4 weeks followed by a decrease of 1.25–2.5 mg per day every 2–6 weeks.

We divided the patients into groups with respect to aetiology, myocardial involvement, treatment and ROP appearance.

Basic (descriptive) statistics included mean values, standard deviations, median and interquartile ranges of monitored parameters. Furthermore, the difference in the distribution of certain features among the tested groups has been determined using the χ^2^ or Fisher’s test. The normality of the distribution of numerical variables had been tested using the Shapiro Wilk and Kolmogorov Smirnov tests. The comparison between the groups was done using the Student t-test, ANOVA, Mann-Whitney test and Kruskal Wallis test. Binominal and multinomial logistic regression analysis was used to explain the relationship between the dependent binary variable and independent variables. All statistical methods were significant if *p* value was ≤0.05. Data processing was done using statistical software SPSS 25.0 for Windows 10.

## 3. Results

In our hospital 68 patients with acute pericarditis were treated between January 2011 and March 2019. The study included 56 patients with idiopathic and viral index attack—41 boys (73.2%) and 15 girls (26.8%). The mean age of patients at the time of the onset of acute pericarditis was 12.31 ± 3.21 years. ROP occurred in eight patients (four after idiopathic and four after viral acute pericarditis) with a mean age of 13.44 ± 2.69 years—12 boys (75%) and 4 girls (25%). The first recurrence occurred in an average period of 6.7 ± 6.3 months after acute disease. The total recorded number of recurrence episodes was 39 (2; IQR: 1–6). The mean time of the achieved remission (period from the last episode of pericarditis to the end of monitoring) was 26.3 months, whereas the maximal period of 48 months without relapses was achieved in three patients.

### 3.1. Aetiology

The aetiology of acute pericarditis and ROP are shown in Table 1. Idiopathic acute pericarditis was significantly the most common diagnosis in our patients. The patient with the largest number of relapses had no specific genetic finding for Familial Mediterranean fever and was classified into the idiopathic recurrent pericarditis group. In the group of patients with viral aetiology the most frequent findings were the Adeno- and Epstein-Barr virus infections. Patients with viral pericarditis had six times more chance of ROP (*n* = 4/6, 66.6%) than those with idiopathic pericarditis (*n* = 4/46, 9%) (*p* = 0.02; OR 6.0; 95% CI = 2.72–36.78) as well as higher median number of recurrence (Figure 1).

### 3.2. Clinical Findings

The difference between patients with idiopathic and viral index attack in laboratory parameters is presented in Table 2. On admission THE mean effusion diameter was 10.52 ± 6.56 mm. Myocardial involvement was registered in 14 patients (8 patients with idiopathic and 6 with viral acute pericarditis). Perimyocarditis was found in 7/14 patients (4 children with idiopathic and 3 with viral index attack). Children with myocardial involvement had higher level of CRP than children with isolated pericardial affection (CRP—160.50 ± 80.52 vs. 113.15 ± 41.14 mg/L, *p* = 0.008). Patients with perimyocarditis had higher values of CRP and ESR than those with myopericarditis (CRP 204.83 ± 73.60 vs. 105.30 ± 65.34 mg/L, *p* = 0.007; ESR 70.71 ± 15.68 vs. 46.43 ± 16.28 mm/h, *p* = 0.02). Additionally, CRP level, ESR and WBC were statistically higher in patients with recurrent pericarditis than with acute disease (CRP—160.57 ± 83.63 vs. 116.62 ± 49.57 mg/L, *p* = 0.05; ESR—72.0 ± 18.35 vs. 47.11 ± 15.67 mm/h, *p* < 0.001; WBC—19.32 ± 5.30 vs. 15.82 ± 3.64 × 10^9^/L, *p* = 0.02).

The average number of WBC was statistically significantly smaller in the first ROP than in index attack (*p* = 0.02).

### 3.3. Treatment

The treatment of acute pericarditis included either monotherapy with NSAID or the combination of NSAID and corticosteroids. An approximately equal number of patients received NSAID monotherapy (29/56 or 51.8%) and a combination of CS and NSAID (27/56 or 48.2%). This drug combination was more often used in patients who were treated before 2015 (*p* = 0.03).

Monotherapy was often used in patients with idiopathic pericarditis (Table 3).

Recurrence of the disease was frequently found in patients treated with a combination of NSAID and CS (6/27, 28.57%) than in patients treated with NSAID only (2/29, 7.41%) (*p* = 0.13). We showed that the children who received a smaller dose of CS (0.5 mg/kg) had smaller chances of recurrence than those treated with high-dose CS (1 mg/kg) (*p* < 0.001). The trigger for recurrence was discontinuation of CS therapy in 5/8 patients. The median number of relapses in the NSAID group was 0 (IQR: 0–0) and 0 (IQR: 0–1) in the CS group (*p* = 0.08). Additionally, CS treatment was associated with a higher recurrence rate (*p* = 0.03) and higher median number of relapses (*p* = 0.01) in children with idiopathic pericarditis, but none in patients with viral index attack (Table 2). CS use in patients with concomitant myocardial affection did not influence the ROP appearance (*p* = 0.38).

Colchicine, NSAID and CS were used for treatment of the first ROP (Table 4). Among the eight patients with relapses, three did not have additional episodes. All patients without additional relapses were treated with a combination of colchicine and NSAID. Children with ROP treated with CS and NSAID had a bigger median number of additional relapses (5, IQR: 2–15) than those treated with colchicine and NSAID (0, IQR: 0–0.75) (*p* = 0.02). Additional relapses in CS patients appeared upon the discontinuation or reduction of steroids. Colchicine was added in therapy after several relapses in two patients treated with a combination of NSAID and CS in the first recurrence. One of them had several ROPs after exclusion of colchicine and the other one has developed CS-depended pericarditis. Adverse events of colchicine (elevated level of transaminases and cytopenia) have not been registered during the follow up period. Only two patients had nausea during the first week of drug intake.

Multinomial logistic regression analysis has shown viral pericarditis (*p* = 0.01, OR 31.46; 95% CI 2.24–441.02), lack of concomitant myocardial affection (*p* = 0.03, OR 29.15; 95% CI = 1.39–609.37), CS use with NSID as treatment of acute pericarditis (*p* = 0.02, OR 29.02; 95% CI = 1.51–557.21) and ESR ≥ 50 mm/h (*p* = 0.03, OR 25.23; 95% CI 1.36–468.86) were independent risk factor for ROP (Table 5).

## 4. Discussion

### 4.1. Aetiology

Thoracic chest pain has a non-cardiac related origin in 97% of paediatrics patients. Regardless, pericarditis is the most common cause of cardiac related chest pain in childhood [10]. In 70% of children, a specific aetiology of acute pericarditis cannot be detected and pericarditis is considered to be idiopathic [4,7,13,14,15]. Idiopathic acute pericarditis is the most common in the United States and Western Europe as well as in our environment [16]. Some investigators presumed that the idiopathic index attack was post-viral [16,17], because the pathogenesis of idiopathic pericarditis was comparable to inflammatory diseases [3]. Brucato et al. claimed that most cases of idiopathic pericarditis were viral in the index attacks, whereas recurrences were often due to a very rapidly tapered drug regimen [3]. Certainly, our patients with idiopathic pericarditis had viral aetiology of the disease but putative viruses were not detected. Contrarily, laboratory analyses were statistically significantly different between children with proven viral and idiopathic index attack. The definitive diagnosis of putative viral agents is dependent on its direct demonstration in the pericardial fluid or tissues but this is not routinely recommended [5]. The pericardiocentesis is routinely preformed in cases of cardiac tamponade, purulent or neoplastic pericarditis [13]. This procedure should be proposed as a second line of investigation when serological tests are negative and in the cases of post-viral pericarditis, which does not resolve despite a longer period of anti-inflammatory treatment [18]. In our study, three patients underwent pericardiocentesis due to haemodynamic reasons. Molecular analysis (PCR) was performed in those children and viral nucleic acids were uniformly negative. Gaaloul et al. showed that PCR in blood and pericardial fluid were simultaneously positive in more than 60% of patients [19]. Other authors claim that the detection of viral nucleic acid in pericardial fluid sample should be interpreted with caution when the agent has not been described previously [18].

Recurrent pericarditis is one of the most important complications of the disease. The etiopathogenesis of the recurrent pericarditis is presumed to be immune-mediated although 80–90% of cases remain idiopathic. Infectious, autoimmune and autoinflammatory pathways have been proposed as mechanisms involved. The interaction between the aetiological agent, genetic factors and the immune system is responsible for the development of the disease. Additionally, the same interactions with potential triggers may precipitate recurrences in predisposed individuals [4]. In our study, patients with viral acute pericarditis had six times larger predisposition for recurrence. Those patients had higher levels of acute phase reactants in the blood. It could be the aftermath of more extensive destruction of the pericardial tissue and expression of more antigens. Consequently, molecular mimicry or superantigens lead to autoinflammatory diseases. Recurrence might be developed due to re-infection [13].

The elevated CRP level was defined as one of the risk factors for unfavourable prognosis after acute pericarditis in adults [16]. In our study, the children with higher level of CRP frequently had ROP as well as the children with higher ESR and WBC. According to our investigation, ESR over 50 mm/h was an independent risk factor for recurrence.

Multinomial logistic regression analysis showed that the lack of myocardial involvement and viral aetiology of the disease were independent risk factors for relapses. An earlier study in adult patients also demonstrated that adult patients with myopericarditis or perimyocarditis had a lower recurrence rate than those with pericarditis only [11].

### 4.2. Treatment

Almost one half of our patients with acute pericarditis ware treated with NSAID (ibuprofen), and the other half received a combination of NSAID and CS. According to the recommendations, patients were treated until clinical and laboratory findings (CRP) were normalized [20]. Some authors recommend that acute pericarditis can be treated with naproxen (10 mg/kg/day), and indomethacin (1–2 mg/kg/day) in severe cases [13]. A double blind study in a single center has shown that ibuprofen and indomethacin had similar efficacy and that both were more effective than placebo [21]. In our investigation, only ibuprofen was used in the index attack. Corticosteroids are the second line of therapy [10] and should be provided for individuals with connective tissue diseases, autoreactive or uremic pericarditis [22]. Imazio et al. administered CS in adulthood only in cases of intolerance, contraindications or an incomplete response to NSAID [11]. We introduced CS in more difficult cases, especially in patients with myocardial involvement, bigger pericardial effusion diameter, suspected autoreactive pericardial inflammation, after pericardial drainage and in children who did not respond adequately to NSAID. Additionaly, CS was administered more frequently in patients who were treated before 2015. Colchicine has been rarely administered in acute pericarditis in childhood. Some authors had been using successfully colchicine in paediatric patients with Familiar Mediterranean fever [23]. In the largest paediatric study, colchicine was used in combination with NSAID and/or CS and did not decrease the readmission rate [23]. However, there is evidence that colchicine (together with NSAID) prevents the relapse of the disease and the duration of symptoms. Hospitalization is not usual in more than 50% of adult patients if it is administrated for three months [13,24,25]. Mager et al. concluded that the use of colchicine in addition to NSAID or aspirin for acute pericarditis in adulthood decreased the recurrence rate by 37%, but without statistical significance [25].

Risks for an unfavourable prognosis after acute pericarditis is defined in adults, but the definition of risks in childhood is lacking. Shakti et al. concluded that the geographic area had borderline association with the recurrence rate [23]. According to Cremer at al. colchicine decreased the chance for recurrence. On the opposite, an incomplete response to NSAID was related with unfavourable prognosis [24]. Missed viral aetiology may contribute to ROP, because corticosteroids could promote viral replication and thus prolong the treatment course [22]. In our study children with viral aetiology of the disease treated with NSAID and CS did not have a higher recurrence rate than those treated only with NSAID (*p* = 0.63).

There is inconsistent data regarding the prognostic role of CS treatment [24,26]. Few studies have shown that CS treatment caries a risk for CS-dependent pericarditis [17], which was supported by our investigation. Corticosteroids can prevent symptoms and suppress inflammation, but their usage can increase the relapse rate by two-fold [13,27,28]. Previously, investigators showed that adults who were treated with high-dose CS (1 mg/kg) had a higher recurrence rate than those treated with 0.2–0.5 mg/kg [29]. Our study confirmed this statement in the paediatric population. Accordingly, children treated with small-dose CS (0.5 mg/kg) had smaller chances of recurrence. Furthermore, we showed that CS treatment increased chances for ROP 29.02 times. Our findings have supported earlier studies in the adult population, which showed that the usage of CS in the treatment of pericarditis increases the frequency of relapses. Additionally, pre-treatment with corticosteroids for the index attack and recurrent effusions may attenuate the efficacy of colchicine in preventing recurrent pericarditis [24,30]. According to a study by the Sackler Faculty of Medicine (Tel Aviv University) the recurrence rate grew when colchicine was administered in combination with CS for the treatment of acute pericarditis. However, CS was not indicated when the CRP level was within normal limits [27]. Assolari et al. suggested that CS should be included in case of autoimmune recurrent pericarditis and incomplete response to NSAID and colchicine, but only in combination with these drugs [31].

In our study, the first relapse of pericarditis was treated with NSAID and colchicine or CS and NSAID. All patients treated with CS and NSAID had additional ROPs over the period of 60 months and only one treated with colchicine and NSAID. A lower number of recurrence was demonstrated in groups of patients treated with colchicine than in the group without colchicine (*p* = 0.02). Few investigations have provided strong evidence for the favourable use of colchicine in adult patients with recurrent pericarditis, while data for paediatric patients is still missing [13,32]. Our results suggest that colchicine in combination with NSAID should be recommended as the treatment of choice for the first recurrence of idiopathic and viral pericarditis in childhood. Additionally, we showed that the treatment of recurrent pericarditis with colchicine was a safe and effective child therapy. However, Raatikka et al. concluded that colchicine did not decrease the number of relapses in children with recurrent pericarditis [26].

Anakinra can be used if recurrent pericarditis is refractory on NSAID, CS and colchicine. The role of anakinra (IL-1 antagonist) is especially important in CS-dependent and colchicine-resistant pericarditis [13,24], antibody negative pericarditis followed by high levels of CRP, high fever, pleuropulmonary and systemic involvement [3]. The efficacy of anakinra was demonstrated in both adults and children, but in a small series and with a potential for recurrence after drug discontinuation [24].

## 5. Conclusions

Independent risk factors for recurrence are viral aetiology, lack of concomitant myocardial affection, CS treatment of acute pericarditis and ESR ≥ 50 mm/h. Idiopathic index attack should be treated with NSAID, while CS use increases chances of ROP. The use of CS in patients with concomitant myocardial affection does not increase the risk for recurrence. However, CS should not be used in first ROP due to higher chances of additional relapses. Combination of colchicine and NSAID might be recommended as the treatment of choice in paediatric patients with the first recurrence of pericarditis. Consequently, the protocols for adults can be applicable for children with viral and idiopathic pericarditis.

### Limitations

The major limitations of this study were collection of data from medical records and retrospective design. The number of patients with viral versus idiopathic pericarditis as well as patients with recurrent pericarditis were insufficient for a final conclusion. Further investigation including higher number of patients should be conducted to confirm our hypothesis that colchicine is drug of choice for treatment of recurrent pericarditis in childhood. Anakinra was not used in our study.

## Figures and Tables

**Figure 1 medicina-55-00609-f001:**
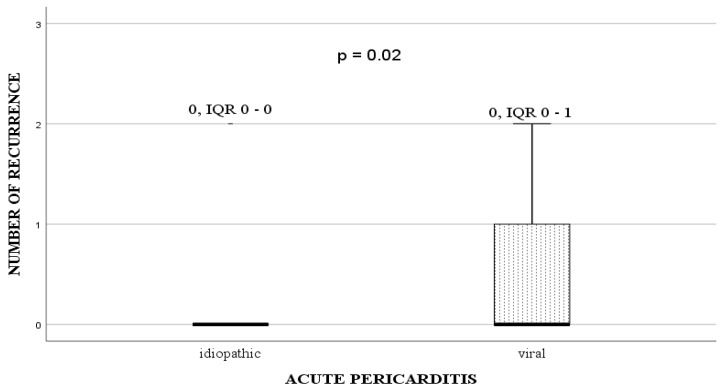
Total number of relapses in patients with acute pericarditis Each box shows the median and interquartile range (the box length). Outliers represent children with values more than 1.5 box lengths from the upper or lower edge of the box Abbreviations: CS—corticosteroids, NSAID—nonsteroidal anti-inflammatory drugs.

**Table 1 medicina-55-00609-t001:** Aetiology of acute and recurrent pericarditis.

Aetiology	Acute Pericarditis-No (%)	First Recurrence-No (%)
Idiopathic	46 (82.14)	4 (50.0)
Viral	10 (17.24)	4 (50.0)
Adeno	3	2
EBV	3	2
Coxsackie Parvo B 19	2	0
	1	0
Influenza	1	0
Total	56 (100)	8 (100)

Abbreviations: EBV—Epstein Barr virus, No—Number.

**Table 2 medicina-55-00609-t002:** Differences in the parameters of acute phase between patients with idiopathic and viral pericarditis.

	All Patients	Idiopathic Pericarditis	Viral Pericarditis	*p* Value
CRP (mg/L)	122.31 ± 56.09	114.64 ± 48.29	160.67 ± 77.66	0.02
ESR (mm/h)	50.73 ± 18.20	48.71 ± 17.36	59.80 ± 20.06	0.08
WBC (× 10^9^)	16.33 ± 4.07	15.52 ± 3.83	19.96 ± 3.12	0.001
Fever (C)	37.47 ± 2.42	37.16 ± 2.52	38.89 ± 1.20	0.04

Abbreviations: CRP—C reactive protein; ESR—Erythrocyte sedimentation rate; WBC—White blood cells.

**Table 3 medicina-55-00609-t003:** Treatment of acute pericarditis regarding aetiology.

Aetiology	Treatment of Acute Pericarditis	Number of Patients (No)	Recurrence (No)	*p*-Value
Idiopathic	NSAID	26	0	0.03
	NSAID + CS	20	4	
Viral	NSAID	3	2	0.5
	NSAID + CS	7	2	

Abbreviations: NSAID—Nonsteroidal anti-inflammatory drugs; CS—Corticosteroids; No—Number.

**Table 4 medicina-55-00609-t004:** Treatment of the first recurrence.

First Recurrence Treatment	No (%) of Patients with First Recurrence	No of Patients with Additional ROPs	No (Median and IQR) of Additional ROPs
NSAID + COLCHICINE	4 (50%)	1/4	0 (IQR: 0–0.75)
NSAID + CORTICOSTEROID	4 (50%)	4/4 *	5 (IQR: 2–15)
Total	8 (100%)	5/8	1 (IQR: 0–2.5)

Abbreviations: NSAID—Nonsteroidal anti-inflammatory drugs; No—Number. * In 2/4 patients colchicine was introduced in additional recurrence of pericarditis (ROPs); six and two attacks respectively.

**Table 5 medicina-55-00609-t005:** Multinomial logistic regression analysis.

	*p* Value	OR	CI
Viral acute pericarditis	0.01	31.46	2.24–441.02
Lack of myocardial involvement	0.03	29.15	1.39–609.37
Treatment (NSAID + CS)	0.02	29.02	1.51–557.21
ESR ≥ 50 mm/L	0.03	25.23	1.36–468.86

Abbreviations: NSAID—Non steroidal anti-inflammatory drugs; CS—Corticosteroids; ESR—Erythrocyte sedimentation rate; OR—Odds ratio; CI—Confidence interval.

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
