# Peer review of "Does Colchicine Substitute Corticosteroids in Treatment of Idiopathic and Viral Pediatric Pericarditis?"

_medicina, 2019, doi:10.3390/medicina55100609_

Round 1
Reviewer 1 Report
While the concept is interesting and important the paper needs a thorough revision. There are entire sections of this paper that are very difficult for the reader to follow. The authors are making very fractured arguments and there is no flow in the paper.
Your study is partly retrospective and there are very few viral pericarditis patients (documented by serologic testing) compared to idiopathic hence statistically teh smallest change in that group will cause statistically significant results. The study is certainly not powered to give conclusive answers. With the patients you have given a range of steroids and a range of NSAIDs, clear tapering parameters are not known and it is unclear if there was a protocol at your institution or were individual practitioners prescribing the medications and tapering doses in an individualized fashion?
There are certainly some good arguments in the paper but they need to be presented in a more cohesive fashion.
Author Response
Dear Reviewer,
Thank you for the suggestion and comments. The corrections had been done according your comments,
as following:
1) Our investigation was checked and corrected by native English speaker;
2) We have made adjustments to the text in order to make it more comprehensible.
3) Differences in the parameters of the acute phase between the patients with idiopathic and viral
pericarditis were presented in Table 2. We have proven that the statistical difference between
two groups exists, but because of the smaller number of patients with viral etiology of the
disease further clinical investigations should be conducted to certify our statement.
4) The therapeutic protocol used in our Institution is better explained. The drugs dosage and
tapering were adopted from the ESC guidelines for treatment of pericardial disease. Prior to
that, the management of the majority of patients with pericardial disease was the
administration of CS in dose of 1 mg/kg. Instead and thereafter we administered CS in a smaller
number of patients, and in smaller doses - 0.5 mg/kg. The biggest difference is that we had only
used colchicine for the recurrence of pericarditis. The dosage and tapering of the drug were the
same as in the ESC protocol. The corticosteroid therapy was discontinued first, and the dose was
decreased after the normalization of clinical and laboratory parameters. If the dose had been
>50 mg/day, it would have been decreased to 10 mg/day every 1–2 weeks; if the dose was 25
and 50 mg/day, it was decreased to 5–10 mg/day every 1–2 weeks; when the administered
dose was between 15 and 25 mg/day, the decrease was by 2.5 mg/day every 2–4 weeks
followed by a decrease of 1.25–2.5 mg per day every 2–6 weeks.
5) In the section “Limitations” we have emphasized that we did not have a sufficient number of
patients with viral pericarditis and patients treated with colchicine. Further investigations
including a higher number of patients should be conducted to confirm our hypothesis that
colchicine is the drug of choice for treatment of recurrent pericarditis in childhood, as well as
the difference between patients with diagnosed viral and idiopathic pericarditis.
With kindest regards
Reviewer 2 Report
The manuscripts presents the clinical experience of the authors with recurrent pericardium in children. The study is well-designed, although it would be limited by the retrospective character as mentioned by the authors. What the authors found as risk factors are yet well known and widely reported in the literature
Author Response
Dear Reviewer,
Thank you for your suggestion and comments.
We hope that we made the corrections according Your comments.
Our investigation was checked and corrected by native English speaker.
With kindest regards.
Reviewer 3 Report
There is lack of evidence to support/discourage use of colchicine in pediatric percarditis. With limited patient population and studies available, I am glad this study adds the experience of a single center to the literature.
Method section:
Line 45: what is the study design? This seems to be a retrospective study. What is meant by prospective cohort study in addition.
Please add the dose and specific duration of colchicine therapy
Result section: Line 161-162 and Discussion treatment section: Line 267-268. P value quoted is 0.02. Since N for total recurrence is 8. Which statistical test was used for this?
Also, two patients out of 4 in NSAID and CS group ended up getting colchicine after recurrence. It would be more useful to make a descriptive table of the patients with recurrence. If not please add in supplement(**) the description of the the patients receiving colchicine after recurrence.
The sample size is small to generalize these results which author's themselves list in discussion( line 281-282)
Author Response
Dear Reviewer,
Thank you for your suggestion and comments.
We hope that we made the corrections will according Your comments, as following:
1) our investigation was checked and corrected by native English speaker;
2) we better explained the therapeutic protocol used in our Institution. The therapeutic protocol used in our Institution was better explained. The drugs dosage and tapering were mostly adopted from the ESC guidelines for treatment of pericardial disease. Prior to them, the management of the majority of patients with pericardial disease was the administration of CS in dose of 1 mg/kg. Instead and thereafter we administered CS in a few of patients, and in smaller doses - 0.5 mg/kg. Colchicine was used only in the first or additional recurrence of pericarditis. The drug dosage and tapering as well as duration of the therapy were adopted from the ESC guidelines for recurrent pericarditis;
3) Mann Whitney test is a non-parametric test used to compare the number of relapses in two different patient groups (Lines 175 – 176 in the section results; Lines 283 - 284 in the section Discussion treatment);
4) We added supplement in the Figure description about patients receiving colchicine after the first ROP. In 2/4 patients colchicine was introduced in additional ROPs; six and two attacks respectively;
5) At the end of the paper in the section “Limitations” we have emphasized that we did not have a sufficient number of patients with viral pericarditis and patients treated with colchicine. Further
investigation including higher number of patients should be conducted to confirm our hypothesis that colchicine is drug of choice for treatment of recurrent pericarditis in childhood.
With kindest regards
Round 2
Reviewer 1 Report
The corrections made reflect the data collection strategy and results with more accuracy.